# Effect of Hydrothermal Cooking and Germination Treatment on Functional and Physicochemical Properties of *Parkia timoriana* Bean Flours: An Underexplored Legume Species of *Parkia* Genera

**DOI:** 10.3390/foods11131822

**Published:** 2022-06-21

**Authors:** Seema Vijay Medhe, Manoj Tukaram Kamble, Aurawan Kringkasemsee Kettawan, Nuntawat Monboonpitak, Aikkarach Kettawan

**Affiliations:** 1Department of Food Chemistry, Institute of Nutrition, Mahidol University, Salaya, Phutthamonthon, Nakhon Pathom 73170, Thailand; seemavijay.med@mahidol.ac.th (S.V.M.); aurawan.kri@mahidol.ac.th (A.K.K.); 2Department of Anatomy, Faculty of Science, Mahidol University, Bangkok 10400, Thailand; maav.manya@gmail.com; 3Department of Food Toxicology, Institute of Nutrition, Mahidol University, Salaya, Phutthamonthon, Nakhon Pathom 73170, Thailand; nuntawat.mon@mahidol.ac.th

**Keywords:** *Parkia timoriana*, germination, hydrothermal cooking, physicochemical properties, functional properties, antioxidant activity, FTIR analysis, microstructure, food security

## Abstract

The present study was undertaken to analyze the impact of germination (NBG) and hydrothermal cooking (NBHTC) on the nutritional profile and physicochemical, functional and microstructural properties of Nitta bean (*Parkia timoriana*) (NBR) seeds. Results demonstrated that the highest crude protein and fat content could be found in NBG and NBHTC, whereas the ash content was significantly higher in NBG. Compared to NBHTC and NBR, NBG has higher emulsion capacity and stability, with values determined to be 58.33 ± 1.67 and 63.89 ± 2.67, respectively. In addition, the highest color intensity was also reported for NBG, followed by NBHTC and NBR. Likewise, NBG showed complete gel formation at a lower concentration (12 g/100 mL) than NBR flour (18 g/100 mL). Furthermore, structural changes in the lipid, protein, and carbohydrate molecules of NBG and NBHTC were evidenced by FTIR studies. Morphological changes were noticed in different samples during microscopic observations subjected to germination and hydrothermal treatment. In contrast to NBR and NBHTC, NBG showed the highest total polyphenol content, ORAC antioxidant, and DPPH radical scavenging activity, which demonstrated the potential utilization of Nitta bean flour as a natural plant-based protein source in food security product formulations.

## 1. Introduction

Food legumes, or “poor man’s meat”, occupy an important place in human nutrition because of their exceptionally high nutritional benefits, rendering them an ideal food worldwide [1]. Major cash crops such as soybean, beans, and peas are being exploited extensively. However, there is still a significant gap in the utilization of some of the underutilized legumes that remain unexplored regarding their nutrition potential and prospective utilization in the food processing industry [2,3]. Parkia, belonging to the family Leguminosae, comprises 30 or more species, of which only a few are traditionally utilized and widely recognized. Nitta bean (*Parkia timoriana*) is a well-known underutilized nutritious leguminous tree found on the Asian continent [4]. It is a valuable tree that ranges from 40 to 820 m a.s.l. in tropical and subtropical zones [5]. The common names of Nitta beans are Burma-Mai-Karien, Thailand-Riang, Malay Peninsula-Kedawong, Kada-ong, Petai kerayong, Indonesia-Alai, India-Supota, Yongchak, and Khorial [6]. Nitta beans are an excellent source of various nutrients such as proteins (globulins and albumins), minerals (iron, potassium, zinc, magnesium, manganese, and phosphorus), amino acids (leucine, isoleucine, tyrosine, and phenylalanine), and fatty acids (linoleic and oleic) [7]. Nitta bean can be eaten after germination and dehulling or cooked dehulled with vegetables or meat [1]. Furthermore, Nitta bean exhibits potent antioxidant and numerous biological activities [6]. It also contains high concentrations of phytochemicals, which play a substantial role in antibacterial, anti-aging, and anticancer activities [8]. 

Despite its nutritional and nutraceutical benefits, antinutritional factors prohibit the direct use of Nitta bean seeds. However, by deactivating antinutritional factors through various industrial-scale processing methods, the digestibility and bioavailability of nutrients can be increased [2]. Nitta bean seeds, before consumption, are conventionally prepared by soaking, roasting, germinating, and cooking methods. Germination has been a potent process for eliminating antinutrient factors and catalyzing secondary metabolites such as phytates and α-galactosides. During the cooking process, antinutrients such as oligosaccharides (causing flatulence) and trypsin inhibitors can be decreased or eliminated [9]. The significant loss of oligosaccharides, phenolic content, minerals, tannins, and phytic acid occurs when legumes are cooked at high temperatures, which is a regularly used processing method for legumes [10]. Thus, germination is an effective method of increasing the antioxidant capacity of legumes [11]. 

Physical characteristics, such as mass, length, width, thickness, geometric mean diameter, surface area, volume, true density, bulk density, porosity, sphericity, and aspect ratio, are critical factors when designing machines for handling, processing, and storing seeds. Many studies have reported the physicochemical and mechanical attributes of different seeds. However, the physicochemical and functional properties of raw, germinated, and hydrothermally cooked Nitta bean flours have not yet been explored. 

The purpose of this study is to highlight the influence of processing on the potential physicochemical and functional properties of Nitta bean flour. Influencing parameters such as emulsion, foaming, thermal characteristics, and other factors have been taken into account in order to study the effect of germination and hydrothermal process on the functional properties of the flours. Since Nitta bean flour has excellent potential as a raw material for food processing industries, it is crucial to examine its physicochemical and functional properties. Information regarding physicochemical parameters is vital for designing equipment for harvesting, storage, transportation, grading, and so on. Thus, the main objective of this study is to assess the physicochemical and functional properties of raw and processed Nitta bean flours to increase their utilization for food product development. To the best of our knowledge, there are not many publications related to the comparative study of the physicochemical and functional properties of raw, germinated, and hydrothermally cooked Nitta bean flours.

## 2. Materials and Methods

### 2.1. Sample Preparation

#### 2.1.1. Raw Nitta Bean (NBR)

Raw Nitta bean seeds were procured from a local market in the Salaya subdistrict (Nakhon Pathom, Thailand). The seeds were manually cleaned to remove dust, debris, and other foreign matter. A Hsiang Tai grinder (Model-SM-3L) was used to grind the raw seeds into flour, then sieved (USA Standard Testing Sieve No. 50). The flour was stored in glass containers (airtight) at −20 °C for further investigation.

#### 2.1.2. Germinated Nitta Bean (NBG)

Prior to germination, the seeds were soaked and manually scarified by secateurs and cutting the seed coat approximately 1 mm from the opposite side of helium [12]. Scarified seeds were rinsed and steeped in distilled water (1:10 *w*/*v*) for 24 h, washed, and placed in a damp cotton towel. The wrapped seeds were placed in the dark at room temperature in an airtight container. The seeds were rinsed every 12 h, and the loosened seed coat was removed. The seeds were germinated for 3 days, then subjected to freeze-drying at −80 °C for 48 h. Lyophilized seeds were powdered using a Philips blender (HR2118/02, Indonesia) and sieved (USA standard testing Sieve No. 50) to get the seed flour. The flour was stored in a glass bottle (airtight) at −20 °C for further analysis.

#### 2.1.3. Hydrothermal Cooking of Nitta Beans (NBHTC)

The top edges of the raw Nitta bean seeds were cut using a cutter, rinsed twice, and soaked for 12 h in distilled water. Afterwards, the seeds were cooked for a minimum cooking time in distilled water (predetermined for seeds soaked for 12 h). Cooked seeds were bloated with tissue paper to remove the excess water on the surface and subjected to freeze-drying at −80 °C for 48 h. Lyophilized seeds were powdered using a Philips blender (HR2118/02, Indonesia) and sieved (USA standard testing sieve No. 50) to obtain the seed flour.

### 2.2. Physicochemical Parameters 

#### 2.2.1. Length, Width and Thickness

A random sample of 100 seeds in triplicate was used to determine the physicochemical properties. A Vernier caliper (Aerospace, Changsha, Hunan, China) was used to measure seed length (L), width (W), and thickness (T) with a 0.05 mm precision. The size of the seeds in the bulk samples was an essential consideration in the analysis.

#### 2.2.2. Equivalent Diameter 

The equivalent diameter of seeds was calculated using the following equation
Dm = (LWT)^1/3^(1)

#### 2.2.3. Sphericity

The sphericity ɸ is calculated by using following equation
(2)$$ɸ\;=\;[{\left(\mathrm{LWT}\right)}^{1/3}/L]\;\times \;100\;$$

#### 2.2.4. Aspect Ratio Is Calculated by Using Equation

The aspect ratio is calculated by using following equation
Ra = W/L(3)

#### 2.2.5. Seed Volume (V) Is Calculated by Using the Formula

The seed volume is calculated by using following equation
V = πB^2^L^2^/6(2L − 3)(4)
where B = (WT)^1/2^

#### 2.2.6. Surface Area Is Calculated by Following Equation

The surface area is calculated by using following equation
V = πBL^2^/2L − B(5)

#### 2.2.7. Seed Weight 

The weight of 100 randomly selected seeds of NBR, NBG, and NBHCT were determined in triplicate. The counted seeds were weighed using a Mettler Toledo (XP2003SDR, Greifensee, Switzerland) with an accuracy of 0.001 mg.

#### 2.2.8. Seed Volume 

The volume of 100 randomly selected NBR seeds was determined in triplicate [13]. In a 500 mL measuring cylinder, the seeds were transferred into 250 mL of distilled water. The difference between the initial and final volumes was noted as the volume of the 100 seeds. The 100-seed volume was divided by 100 to obtain the volume per seed.

#### 2.2.9. Husk Content

The seeds (10 g) were steeped in DW (50 mL) for 24 h at 25 °C. Seed coats were manually removed. The cotyledons and seed coats were dried separately for 12 h at 70 °C, followed by cooling at 25 °C for an hour. The husk content was calculated as the percentage of the weight of the seed coat to the whole seed weight.

### 2.3. Field Emission–Scanning Electron Microscopic (FE-SEM) Analysis

The morphologies of NBR, NBG, and NBHTC were examined using a scanning electron microscope (JSM-7610FPlus, Freising, Germany) with an acceleration voltage of 5 kV [14]. Before coating, the lyophilized samples were mounted on the specimen stub using double-sided tape. Platinum sputtering was used to coat the sample. The moisture content of lyophilized samples prepared for FE-SEM was lower than 1% for all samples.

### 2.4. Proximate Composition

The moisture (AOAC 931.04, Oven—Memmert UN-110, Schwabach, Germany), crude fat (AOAC 922.06, Soxhlet—Soxtec GR-47 RA-13 006, Höganäs, Sweden), ash (AOAC 930.30, Muffle Furnace—Carbolite CWF1100, Hope Valley, England), crude protein (AOAC 992.23, digestion unit Buchi K435 and distillation unit Buchi B-324, Essen, Germany), total dietary fiber (AOAC 985.29, aspirator pump—Eyela A-1000S, Shanghai, China, oven Memmert UN-1100), and soluble (AOAC 993.19) and insoluble (AOAC 991.42) dietary fiber contents of the NBR, NBG, and NBHTC flours were estimated using the AOAC (2019) official methods [15]. The difference method was used to calculate the amount of carbohydrates in Nitta beans.
Carbohydrates (%) = [100 − Moisture (%) − Proteins (%) − Fats (%) − Ash (%)](6)

### 2.5. Functional Properties

#### 2.5.1. Color Measurement

Color analysis of the seeds and flours was conducted by color spectrometer (ColorFlex EZ, HunterLab, Reston, VA, USA) considering different color scales (CIE L*, a*, and b*). The calorimeter was calibrated with the standard white and black plate. L* denotes lightness [from 0 (black) to 100 (white)], a* denotes reddish (+a*) and greenish (−a*) colors, and b* denotes yellowish (+b*) and bluish (−b*) colors. The measurements were performed under similar light conditions, at room temperature, replicated three times for each flour and bean sample. 

#### 2.5.2. Least Gelation Concentration 

The least gelation concentration was determined using the method described by Medhe et al. [14]. All flour samples were dispersed in 3 mL of distilled water at 2–20% (g/100 mL) in test tubes, followed by heating for one hour in a water bath at 95–100 °C. The dispersion was cooled to 4 °C. The lowest gelation concentration was assessed by visual inspection for slipping out any drops from the emulsion after inverting the tubes. The results are presented as no (−), complete (+), or partial (±) gelation.

#### 2.5.3. Emulsion Capacity (EC) and Emulsion Stability (ES) 

The EC and ES were assessed using the protocol described by Medhe et al. [14]. Flour suspension (100 mL, 5% (*w*/*v*)) was homogenized using an (IKA T25 digital Ultra-Turrax, Shanghai, China) homogenizer at 24,000 rpm for 2 min, followed by the addition of soybean oil (100 mL of the density of 0.912 g mL^−1^) to each sample and homogenization for another 2 min. The emulsion was separated (10 mL) and centrifuged (1048 *g* for 5 min). The emulsion volume was then measured. The emulsion activity was denoted as the percentage of the emulsified layer volume in the entire centrifuge tube solution. To determine the emulsion stability, the prepared emulsions were subjected to heating at 80 °C for 30 min, cooled to room temperature, and centrifuged at 1200 *g* for 5 min. The emulsion stability is presented as a percentage of the remaining emulsified layer volume relative to the original emulsion volume.

#### 2.5.4. Swelling Capacity

Seeds (20 g) were counted in triplicate. The volume was measured, followed by overnight soaking of seeds in distilled water. The volume of the soaked seeds was recorded in a graduated cylinder.
Swelling capacity (mL/seed) = (V_2_ − V_1_)/N(7)
where V_1_ = seed volume before soaking, V_2_ = seed volume after soaking, and N = seed count
Swelling index (SI) = swelling capacity of seed/one seed volume(8)

#### 2.5.5. Hydration Capacity

The seeds (20 g) were counted in triplicate, transferred to a measuring cylinder of 100 mL of distilled water, and left for 24 h at room temperature (28 ± 2 °C). The water was discarded, and the seeds were blotted to remove the adhered water and weighed [13].
Hydration capacity (g/seed) = (W_2_ − W_1_)/N(9)
where W_1_ = seeds weight before soaking, W_2_ = seeds weight after soaking, and N = Seed count.
Hydration index = hydration capacity per seed/weight of one seed(10)

#### 2.5.6. Water Holding and Oil Holding Capacity

Each flour sample (50 mg) was weighed into a pre-weighed Eppendorf tube, followed by the addition of 1 mL distilled water and soybean oil. Vortexed samples were allowed to stand for 30 min at 25 °C before being centrifuged for 25 min at 1048 *g*. The tubes were inverted on absorbent paper to remove the excess water and oil. The water-and oil-holding capacities were calculated based on the difference.

### 2.6. Fourier-Transform Infrared Spectroscopy (FTIR) Analysis

Fourier-transform infrared (FTIR) spectra of the raw, germinated, and hydrothermally cooked Nitta bean flours were obtained using an FTIR spectrometer (Nicolet Summit Pro, Thermo Scientific, Waltham, MA, USA) [16]. The analysis was carried out by mixing Nitta bean flour and KBr in a mass ratio of 1:100, then grinding and pressing the mixture into a pellet. Measurements were obtained in the range of 500–4000 cm^−1^ with a resolution of 4 cm^−1^, and a total of 64 scans were recorded and averaged. The experiments for FTIR were performed in triplicate.

### 2.7. Thermal Properties

Differential scanning calorimetry (DSC) was used to evaluate the Nitta beans’ thermal characteristics. Approximately 20 mg of bean flour sample with 20 µL of water was placed inside a hermetic aluminum pan. The pan was sealed and stabilized at 20 °C for 1 h. Sample heating was performed on an empty pan from 20 to 115 °C at 2.5 °C/min in a DSC chamber with an N_2_ atmosphere (50 mL/min). Universal Analysis 2000 software was used to calibrate the enthalpy of the bean flour.

### 2.8. Antioxidant Activity

#### 2.8.1. Total Polyphenol Content

The total polyphenol content was determined using the Folin–Ciocalteu reagent method, with slight modifications [17]. The flour sample (0.2 g) was extracted with 80% methanol (4 mL) for 2 h in a water bath (shaking) at 30 °C, followed by 10 min centrifugation (2000 *g*). Then, distilled water (500 µL) and extract (10 µL) were mixed in the test tube, and 50 µL of reaction-initiated mixture, i.e., Folin–Ciocalteu reagent (Sigma Aldrich, St Louis, MO, USA) was quickly added. After 3 min of incubation, 200 µL of Na_2_CO_3_ (20 g/L (*w*/*v*)) and 245 µL of distilled water were added to the reaction mixture. The absorbance of the test solution was measured using a spectrophotometer. Gallic acid (10–80 ug/mL) was used as standard. The total phenols were expressed as mg/100 g gallic acid equivalent using the standard curve equation: y = 0.0063x + 0.0245, R^2^ = 0.9993; where y is absorbance at 750 nm, and x is total phenolic content standard.

#### 2.8.2. DPPH Radical Scavenging Activity

The 1,1-diphenyl-2-picrylhydrazyl (DPPH) radical technique was used to assess antioxidant potential [18]. Methanolic extract solution 2 mL was mixed with 2 mL of DPPH solution (0.15 mM, 95% methanol) and incubated at room temperature for 30 min in the dark. The absorbance of the Nitta bean samples and Trolox (a standard) was recorded at 517 nm. The flour’s ability to scavenge free radicals was measured as a mM Trolox Equivalent (TE)/g sample. The analysis was performed in triplicate.

#### 2.8.3. Oxygen Radical Absorbance Capacity (ORAC) Assay

ORAC analysis was carried out using the method of Alberto et al. [18]. Briefly, 20 µL of the sample was placed in a microplate (96-well). The plate was sealed with parafilm and incubated at 38 °C for 30 min in a fluorometer (FLOU Star OPTIMA Microplate Reader), followed by an additional 10 min incubation after uncovering. Later, 200 µL fluorescent solution and 20 µL 3.2 mM 2,2’-Azobis(2-amidinopropane) dihydrochloride (AAPH) were added to all wells. Fluorescence changes at 485 nm (excitation) and 520 nm (emission) were used to determine kinetics. The final ORAC values were determined by plotting the linear regression curve of the Trolox standard or the sample against the region beneath the fluorescence decay curve. The findings are presented in terms of micromoles of Trolox equivalent per gram sample (µM TE/g).

### 2.9. Statistical Analysis

The physicochemical parameters (*n* = 100), proximate composition (*n* = 3), functional properties (*n* = 3), total polyphenol content and antioxidant activity (*n* = 3) were performed in triplicate, and the results are presented in terms of mean ± standard error. The data were statistically analyzed using SPSS software (SPSS Inc. version 26, Chicago, IL, USA), and Tukey’s HSD multiple range test was used to determine significant differences (*p* < 0.05) between the mean values.

## 3. Results and Discussion

### 3.1. Physicochemical Properties

The physicochemical properties of NBR differed significantly from those of NBG and NBHTC seeds (Table 1). The length of NBG (23.75 mm) was noticeably higher than those of NBHTC (19.86 mm) and NBR (17.39 mm), because cooking and germination processes increase the length of seeds as processing ads water absorption. 

The mean values of the length, width, and thickness of the NBR seeds were found to be very similar to the findings reported by Gupta et al. [19]. The maximum increase in width and thickness of seeds was observed during the germination process. Germination increases the porosity of cell walls, reduces the compact intracellular environment, and modifies the macrostructures of proteins, which may help to later absorb water from their matrix more easily [20]. The soaking and cooking processes also enhanced porosity and increased the rate of water absorption. In contrast to NBR, the hydrothermal treatment affected the length, thickness, and diameter of beans.

The volume and surface area of NBG was found to be significantly higher than NBHTC and NBR (Table 1). Nevertheless, the weight of NBR was found to be notably lower than that of NBG and NBHTC. The seed weight and volume of beans increased after germination and hydrothermal treatment; similar findings were reported by Miceli and Miceli [21]. Shapes play a vital role in heat and mass transfer, as it is essential to screen for impurities and assess food quality. The sphericity and aspect ratio of food materials commonly describe their shape. The NBR (54.99%) exhibited the maximum sphericity, while the least was observed in NBG seeds (50.77%). Moreover, our findings are similar to those in the study reported by Falade and Akinrinde [22]. The shape of the seed can be predicted by examining the aspect ratio [23], which varied significantly in the NBG (0.47) compared to the NBR (0.53) and NBHTC seeds (0.53). The content of the husk in NBR was found to be 37.90%, higher than in a previous study, which reported the value to be in the range of 9.27–9.72% [23]. The husk content was absent in the NBG and NBHTC because it was removed during the germination process and after cooking. The physicochemical parameters like length, width, diameter, and thickness were higher for NBG and NBHTC than the NBR.

### 3.2. Proximate Composition

Food processing usually involves a change in the temperature and moisture content, which eventually alters the chemical characteristics of the product. In the food industry, macro- and micronutrients are essential for developing products and quality control. The proximate composition of the NBR, NBG and NBHTC flours is listed in Table 2. 

The moisture content was significantly higher in NBR (8.87%) as compared to the NBG (6.66%) and NBHTC (0.98%). The difference in moisture content could be due to the lyophilization of NBG and NBHTC. NBG (37.34 g 100 g^−1^) and NBHTC (36.63 g 100 g^−1^) found considerably more protein than NBR (17.27 g 100 g^−1^). The fat content was found significantly higher in NBHTC (24.02 g 100 g^−1^) followed by NBG (15.57 g 100 g^−1^) and NBR (8.72 g 100 g^−1^). The protein values in raw seeds are mainly assumed due to the stored proteins in the tissues of seed cotyledon [24]. The hydrolysis of protein during the germination process resulted in an increased protein content. Proteins deposited in the embryonic axis and testas of legume seeds promote enzymatic activity and growth of new tissues during germination, thereby increasing water-soluble proteins [25]. After germination, a further increase in protein content might be due to the enzymatic protein synthesis that resulted in an abundance of protein [26]. Removal of heat-sensitive protein inhibitors and protein denaturation, which causes the globulin structures to open up and make them more accessible, might be responsible for the boosting of the protein content in soaked-cooked Nitta bean seeds. Another possible reason could be the depletion of tannins and phytic acid compounds during the soaking-cooking process [2]. These findings are consistent with the other studies in the literature [25,27].

Compared with NBR, NBG has significantly higher total fat content, which is in association with Borek et al. [28]. The enhanced fat content in NBG might be due to the increase in complex lipids such as phospholipids due to the consequent rise in free fatty acids and depletion of diglycerol during germination [25]. 

NBHTC manifested a decrease in ash content [27], which could be ascribed to the extraction of macro-and micronutrients during soaking and cooking [14]. Ash content can also be reduced by mineral migration into the water during cooking [27,29]. Nevertheless, NBG flour displayed high ash content values, which could be due to phytase activity stimulation, resulting in the hydrolysis of enzymes, proteins, and mineral bonds [17]. The results agree with a previous investigation documenting that germination enhanced the ash content of lentils, soybeans, and chickpeas [25].

Moreover, the total dietary fiber content was significantly higher in NBR (56.52 g 100 g^−1^), followed by NBHTC (32.49 g 100 g^−1^) and NBG (26.86 g 100 g^−1^). Numerous investigations have demonstrated that germination and cooking procedures significantly affect the dietary fiber fractions of various legumes [30,31,32]. Nevertheless, these variations are a legume- and processing-condition specific.

Interestingly, the soluble fiber content was significantly higher in NBR (12.70 g 100 g^−1^), followed by NBHTC (7.81 g 100 g^−1^), and then NBG (1.47 g 100 g^−1^). Comparable results have been reported for soybean [30,33]. During the germination process, the seed coat was removed, leading to a lower total dietary fiber (TDF) in the flour [33], and thus reduced TDF values in NBG. The decreased fiber content of NBHTC flour may be due to the discarding of gluey water after soaking and cooking the Nitta beans and removing the seed coat after cooking. When Nitta beans are cooked, the cooking water became sticky, resulting in a thick syrup that may contain soluble fibers as well as macro- and micronutrients from the Nitta bean. The cooking process softens plant tissues, promotes polysaccharide depolymerization, and increases the water solubility of its fibers [34].

Importantly, the carbohydrate content in NBG (40.78 g 100 g^−1^) was significantly lower than NBR (68.86 g 100 g^−1^) due to the utilization of carbohydrates as substrates for energy generation during the germination process [25]. These results are in agreement with findings for germinated chickpea seeds [35] and desi kabuli chana [36]

### 3.3. Functional Properties

#### 3.3.1. Hunter Color Properties

The Hunter color values of NBR, NBG, and NBHTC flours and seeds are presented in Table 3. NBGF (48.47) showed a substantially higher degree of lightness compared to NBHTCF (43.84) and NBRF (41.84). NBRF reported the lowest degree of lightness, which indicates that the darker color of the flour may be due to the presence of tannins, phytonutrients, and anthocyanin pigments in the seed coat [37]. NBGF revealed higher L values and higher degrees of greenness attributed to the detachment of the seed coat during germination, as well as germination induced multiplication of cells, development of radicles, and embryo shoots [14]. 

The green color of cotyledon also causes the higher L values of NBGF. Enhanced degrees of greenness could also be due to the oxidation of pigmented compounds during germination process. A similar pattern was reported for the L values of the NBR, NBG, and NBHTC seeds. NBRF showed significantly higher a* values (1.59) than NBHTCF, which indicates that the highest degree of redness of the flour could be due to the darker seed coat and varied redness in NBHTCF because of the removal of the seed coat. Importantly, a similar trend in a* values was observed for NBR, NBG, and NBHTC. The highest values of Croma were recorded with NBGF (17.65), followed by NBHTCF (15.78) and NBRF (10.98). Similarly, NBG (30.96) seeds had higher color intensity compared to NBHTC (21.56) and NBR (4.35) seeds.

#### 3.3.2. Least Gelation Concentration 

The gelation properties observed for the NBR, NBG and NBHTC flours at different concentrations (2–20 g 100 mL^−1^) are listed in Table 4. It was observed that the gelation in all samples generally initiated at ≥10 g 100 mL^−1^. Furthermore, NBRF exhibited partial and complete gelation at 14 and 18 g 100 mL^−1^ flour concentration. Notably, the NBG flour showed complete gel formation at 12 g 100 mL^−1^, indicating that the germination process decreased the gelation concentration. This may also imply that the amylase produced during germination interacts with the starch molecules of flour, enhancing its gelation properties. Gelation was also appropriated to the high globulin fraction present in the bean flour. The observed variation in gelling characteristics of flours could be ascribed to the relative proportions of proteins, carbohydrates, and fats, implying that the interaction between these moieties plays a significant role in their functional attributes [38]. Interestingly, NBHTC flour did not form a complete gel. Increased protein concentration enhances the interaction of binding forces, thereby increasing the gelling ability of flour [39]. Compared with NBR, NBHTC showed the higher protein concentration, and least gel-forming ability, which is inconsistent with the findings reported earlier.

#### 3.3.3. Emulsion Properties, Swelling Properties, Hydration Properties, and Water and Oil Holding Capacities

The flour’s emulsion, swelling, hydration, and water- and oil-holding capacity properties are listed in Table 5. The NBG flour demonstrated significantly (*p* < 0.05) higher emulsion capacity (EC) and emulsion stability (ES) than NBR flour, and similar findings were reported for germinated cowpea flour [38]. The enhanced EC and ES of NBG could result from germination-induced polypeptide chain dissociation and fractional unfolding, revealing the hydrophobic sites of amino acids, which later form a complex network with lipids resulting in an increased emulsion capacity. The germination of legumes enhances emulsion stability, which could be attributed to the ability of proteins to act as food surface-active factors. Denatured proteins cause electrostatic aversion on oil droplet surfaces, thereby enhancing the emulsion stability [17,40]. 

Moreover, the increase in emulsifying capacity could also be associated with the germinated flours’ high protein and fat composition [38]. Interestingly, NBHTC flour showed significantly lower EC and ES than NBR and NBG flours, and similar results have been reported for soaked-cooked dehydrated chickpeas [41]. NBR and NBHTC flours could not form foam, whereas NBG flour was found to have 4% foaming capacity with zero foaming stability. The swelling capacity (SC) and swelling index (SI) of NBR, NBG, and NBHTC flours were not significant, which could be attributed to the hard-to-cook characteristics of the Nitta bean. 

The water holding capacities (WHC) of NBR, NBG, and NBHTC flours did not vary significantly (*p* > 0.05), and ranged from 0.81 to 0.83 mL g^−1^. The current results contradict the findings of previous study [41]. The WHC of flour is associated with the denaturation and unfolding of proteins, revealing previously hidden peptide bonds and polar side chains that allow for the storage of additional water molecules. Carbohydrate content includes starch that gelatinizes and dietary fiber that absorbs moisture. Moreover, the oil-holding capacity (OHC) of NBR, NBG, and NBHTC flours was not found to vary significantly (*p* > 0.05), ranging from 0.80 to 0.86 g g^−1^. Aguilera et al. [41] reported that the increase in insoluble fiber content due to processing was attributed to the increased WHC of soaked, cooked chickpea and lentil flour. In the present study, no significant difference in the WHC of NBR, NBG, and NBHTC flours was observed due to the decreased insoluble fiber content after processing.

### 3.4. FTIR Analysis

Figure 1 displays the FTIR spectra of NBR, NBG, and NBHTC in the 500 to 4000 cm^−1^ regions (Table 6). The characteristic absorption band in the studied spectral range (3100–3700 cm^−1^) corresponds to the stretching vibrations of OH in NBR, NBG, and NBHTC, respectively. The increased absorbance was recorded for NBG and NBHTC (3388 cm^−1^). Similarly, absorption peaks appeared in the range (3050–2800 cm^−1^), evidencing the presence of vibrations produced by the asymmetric and symmetric CH stretching of triglycerides (lipid compounds) in various samples. 

It was observed from the spectra that the absorbance recorded a higher value of CH stretch in the case of NBG and NBHTC (2925 cm^−1^ and 2854 cm^−1^) compared to NBR. In addition, the amide I and amide II bands of protein in samples appeared in a spectral range between 1750–1550 cm^−1^ [16]. The C=O stretching vibration of the peptide group produces amide I bands [42], and the CO-NH stretch is responsible for the amide II bands [25]. By analyzing the absorbance spectra, it can be found that the absorbance for the protein content recorded a higher value in NBG and NBHTC than NBR, which is in agreement with previous research [25]. 

The bands in the lower frequency region of 1200–900 cm^−1^ correspond to carbohydrates’ presence in the samples [25,43]. The FTIR results revealed the comparative proximate composition data of the Nitta bean, which indicated that the processed Nitta bean had significantly increased protein content. The band at 1075 cm^−1^ attributed to the vibration of the C-O groups of the carbohydrate recorded an increased intensity in NBG. The results obtained from FTIR corroborated the proximate composition of the NBG, indicating that germination can reduce the total carbohydrate content of Nitta beans. The present results agree with those of reported study [25].

### 3.5. Thermal Properties

The thermal properties of the Nitta bean flours were assessed using DSC, and the resulting transition temperature alterations are shown in Figure 2. The first and second peaks originated were due to the gelatinization of starch and decomposition of amylose-lipid complexes [44]. Germination and hydrothermal cooking are responsible for the significant differences in the onset and peak temperatures of Nitta bean flour. In contrast to NBG and NBHTC, the higher onset temperature of NBR could be due to the higher amount of starch present in it. The highest peak temperature in NBR indicated the increased resistance of the starch towards gelatinization [14].

The transition onset temperature indicated that initiation of starch gelatinization first began in NBG, followed by NBR and NBHTC. NBR had the highest transition peak temperature. The transition peak temperature is strongly related to the amino acid makeup, protein structure, and conformation. As the heating increased, the gelatinization of larger starch molecules started in the NBR flour, followed by NBHTC and NBG. The NBR has the highest conclusion temperature. These results are contrary to the findings reported in previous studies [14,45]. The amylose content, lipid complex, amylopectin chain distribution, and protein content are responsible for the differential thermal profile of bean flour [46]. 

### 3.6. FE-SEM Analysis

The morphologies of NBR, NBG, and NBHTC flours were examined by FE-SEM analysis. In NBR, seed cells and cell walls in the middle separating lamellae were apparent in every scan (×500, ×1000, and ×2000) (Figure 3a–c). The proteins and lipids embedded in the soluble and insoluble dietary fiber structures were clearly visible in NBR. A similar structural morphology has been observed in chickpea and lupin bean flours [25].

The morphology of NBG flour (Figure 3d–f) differed slightly from NBR because of the increased protein and lipid content after the germination process. After seed germination, noticeable structural changes were observed in the cotyledon cells. As germination progresses, the starch and protein contents of the seed change. Germination increases the hydrolytic activity, resulting in cell rupture. Some precipitates have been observed on starch, most likely because of increased proteolytic activity due to germination [47]. Many surface cracks are easily noticeable. The structural changes in the germinated Nitta bean flour were most likely due to the higher protein and lower carbohydrate contents. In the stable cell matrix, which emerged to be composed of compressed materials, spherical and oval protein bodies were visible in the cell. The cells became more sparsely packed following germination, with significant inter-cell spaces. The protein structure appeared to be granular, with multiple cracks occurring due to seed germination, barely visible in the NBR. 

Morphological differences observed in the NBHTC flour (Figure 3g–i) could be attributed to the increase in the protein and lipid composition of the flour after the soaking and cooking processes. More spherical and oval particles embedded in the extracellular spaces of the fibers were observed. The increased soluble and insoluble dietary fiber content made the structure smoother. In addition, the resulted modification could be due to alteration in the structures of the proteins and lipid molecules.

### 3.7. Total Polyphenol Content and Antioxidant Activity

The total polyphenol content (6.88 mg GE/g), ORAC antioxidant activity (505.33 µM TE/g), and DPPH radical scavenging activity (61.21 mM TE/g) of the NBG were significantly higher than those of NBR and NBHTC (Table 7). NBG had the highest polyphenol content, and is responsible for its high antioxidant capacity. These findings are similar to those reported by Pal et al. [48]. Amylases, proteases, and other hydrolytic enzymes generated during germination aids in the release of bound phenolic chemicals [49]. The increased polyphenol content could be due to the elevated activity of the enzyme phenylalanine ammonia-lyase in the course of germination [50]. Hydrothermal cooking decreased the ORAC antioxidant activity, DPPH radical scavenging activity, and polyphenol content of Nitta bean flour. These findings are contrary to those documented by Chipurura et al. [51]. Soaking and thermal treatment cause the softening and disintegration of the cell walls, which leads to the release of extractable phenolics. The lowest phenolic and antioxidant activities were due to the loss of phenolic compounds through discarded water used for soaking and cooking [52]. Hence, increased polyphenol content could increase the antioxidant activity of ORAC and DPPH.

## 4. Conclusions

The current study found that conventional processing techniques, such as germination and hydrothermal cooking, can positively modify the Nitta bean flour’s chemical constituents, functional properties, and antioxidant capacity. Germinated Nitta bean flour exhibited the highest antioxidant capacity, emulsion properties, and the lowest gelation concentration. The germination of seeds resulted in an increased protein and ash content than the rest of the samples, with values noted to be 37.34 ± 0.24 and 6.32 ± 0.03 g 100 g^−1^, respectively. FTIR spectra clearly showed the variation of protein, carbohydrate, and lipids in various samples. The SEM analysis concluded that germination increases the hydrolytic activity, resulting in cell rupture. The noticeable morphological changes in NBG were most likely due to the higher protein and lower carbohydrate contents that are barely visible in the NBR. In addition, NBG demonstrated the most increased DPPH radical scavenging activity, ORAC antioxidant activity, and total polyphenolic content of 61.21 mg GE/g, 505.33 µM TE/g, and 6.88 mM TE/g, respectively. Furthermore, protein isolation and quality analyses are recommended for NBG. The results of this study provide a basis for the exploitation and utilization of Nitta bean flour in product formulations and as a potential source of natural plant-based protein.

## Figures and Tables

**Figure 1 foods-11-01822-f001:**
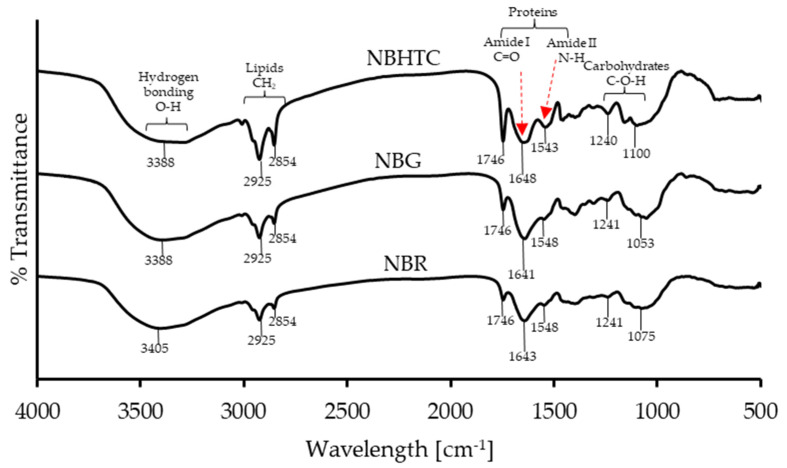
Fourier transform infrared spectra of Nitta bean raw, germinated, and hydrothermally cooked flours.

**Figure 2 foods-11-01822-f002:**
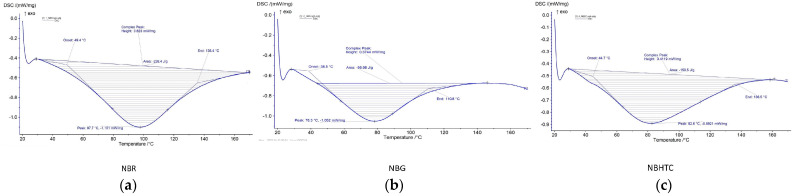
Thermal properties of raw (**a**), germinated (**b**), and hydrothermally cooked (**c**) Nitta bean flour.

**Figure 3 foods-11-01822-f003:**
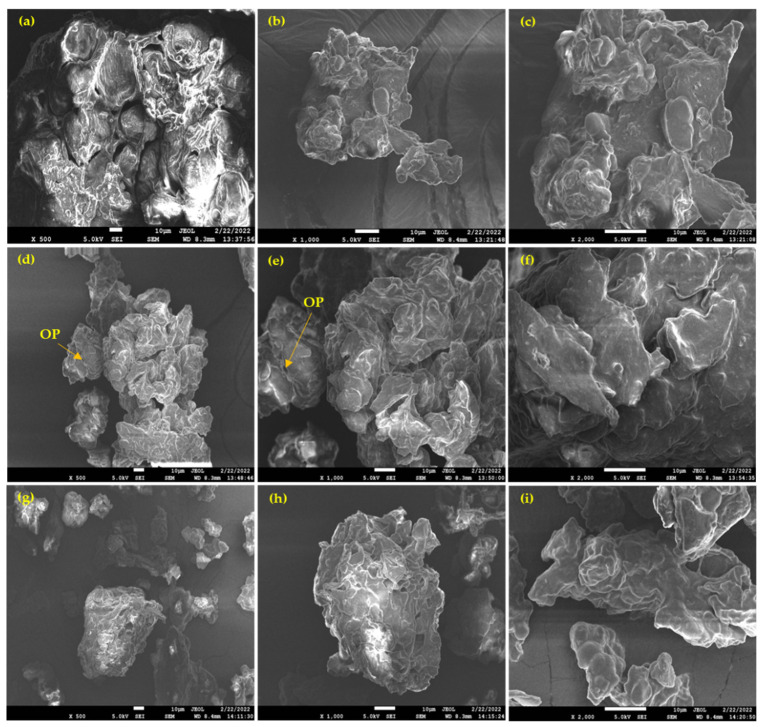
Morphology of raw (**a**–**c**), germinated (**d**–**f**), and hydrothermally cooked (**g**–**i**) Nitta bean flours. OP—oval shape protein.

**Table 1 foods-11-01822-t001:** Physicochemical properties of raw, germinated and hydrothermally cooked Nitta bean seeds.

Parameter	NBR	NBG	NBHTC
Length (mm)	17.39 ± 0.14 ^a^	23.75 ± 0.17 ^c^	19.86 ± 0.16 ^b^
Width (mm)	9.14 ± 0.10 ^a^	11.08 ± 0.12 ^b^	10.67 ± 0.11 ^b^
Thickness (mm)	5.53 ± 0.07 ^b^	6.69 ± 0.07 ^c^	5.09 ± 0.04 ^a^
Diameter (mm)	9.53 ± 0.08 ^a^	12.03 ± 0.08 ^c^	10.22 ± 0.07 ^b^
Sphericity (%)	54.99 ± 0.36 ^b^	50.77 ± 0.27 ^a^	51.61 ± 0.22 ^a^
Aspect ratio	0.53 ± 0.01 ^b^	0.47 ± 0.00 ^a^	0.54 ±0.01 ^b^
Volume (mm^3^)	252.19 ± 5.53 ^a^	491.72 ± 9.07 ^c^	306.50 ± 6.11 ^b^
Surface area (mm^2^)	243.36 ± 4.81 ^a^	391.14 ± 3.65 ^c^	281.88 ± 3.89 ^b^
Weight (g/100 seed)	54.31 ± 0.28 ^a^	94.19 ± 0.65 ^c^	69.18 ± 2.03 ^b^
Volume (mL/100 seed)	43.67 ± 0.88 ^a^	89.00 ± 3.06 ^c^	65.33 ± 3.93 ^b^
Husk content (%)	37.90 ± 0.48	--	--

Results are the means of three replicates ± SE; Tukey’s HSD multiple comparison test, mean values with the same row followed by the same superscripts are not significantly different (*p* > 0.05). --: Not available; NBR—Nitta bean raw; NBG—Nitta bean germinated; NBHTC—Nitta bean hydrothermally cooked.

**Table 2 foods-11-01822-t002:** The proximate composition of raw, germinated and hydrothermally cooked Nitta bean flours (g 100 g^−1^) on dry weight basis.

Parameter	NBR	NBG	NBHTC
Energy (Kcal)	385.51± 0.58 ^a^	422.66 ± 0.20 ^b^	502.37 ± 0.65 ^c^
Moisture	8.87 ± 0.21 ^c^	6.61 ± 0.03 ^b^	0.98 ± 0.02 ^a^
Total proteins	17.27 ± 0.01 ^a^	37.34 ± 0.24 ^b^	36.62 ± 0.02 ^c^
Total fats	8.72 ±0.06 ^a^	15.57 ± 0.02 ^b^	24.02 ± 0.08 ^c^
Total carbohydrates *	68.86 ± 0.29 ^c^	40.78 ± 0.25 ^b^	36.18 ± 0.07 ^a^
Total dietary fibers	56.52 ± 0.44 ^c^	26.86 ± 0.22 ^a^	32.49 ± 0.24 ^b^
Soluble dietary fibers	12.70 ± 0.56 ^c^	1.47 ± 0.02 ^a^	7.81 ± 0.26 ^b^
Insoluble dietary fibers	43.82 ± 0.98 ^b^	25.39 ± 0.22 ^a^	24.67 ± 0.31 ^a^
Ash	5.15 ± 0.03 ^b^	6.32 ± 0.03 ^c^	3.18 ± 0.02 ^a^

Results are the means of three replicates ± SE; Tukey’s HSD multiple comparison test, mean values with the same row followed by the same superscripts are not significantly different (*p* > 0.05). * Value calculated using the difference. NBR—Nitta bean raw; NBG—Nitta bean germinated; NBHTC—Nitta bean hydrothermally cooked.

**Table 3 foods-11-01822-t003:** The hunter color values of Nitta bean raw, germinated and hydrothermally cooked seeds and their respective flours.

Nitta Bean	Hunter Color Values				
	L	a*	b*	Croma	Hue Angle
NBRF	41.84 ± 0.10 ^a^	1.59 ± 0.01 ^c^	10.98 ± 0.01 ^a^	11.09 ± 0.01 ^a^	101.11 ± 0.03 ^b^
NBGF	48.47 ± 0.01 ^c^	−3.47 ± 0.01 ^a^	17.65 ± 0.02 ^c^	17.99 ± 0.03 ^c^	101.11 ± 0.03 ^b^
NBHTCF	43.84 ± 0.06 ^b^	0.20 ± 0.01 ^b^	15.78 ± 0.01 ^b^	15.78 ± 0.01 ^b^	89.29 ± 0.04 ^a^
NBR	21.62 ± 0.04 ^A^	2.37 ± 0.05 ^B^	3.64 ± 0.00 ^A^	4.35 ± 0.03 ^A^	56.90 ± 0.58 ^A^
NBG	56.107 ± 0.07 ^C^	−4.67 ± 0.07 ^A^	30.60 ± 0.07 ^C^	30.96 ± 0.07 ^C^	98.68 ± 0.13 ^C^
NBHTC	35.96 ± 0.17 ^B^	4.53 ± 0.04 ^C^	21.08 ± 0.12 ^B^	21.56 ± 0.12 ^B^	77.86 ± 0.06 ^B^

Results are the means of three replicates ± SE; Tukey’s HSD multiple comparison test, mean values with the same column followed by the same superscripts (small letter for flour and capital letters for seed) are not significantly different (*p* > 0.05). NBRF—Nitta bean raw flour; NBGF—Nitta bean germinated flour; NBHTCF—Nitta bean hydrothermally cooked flour. NBR—Nitta bean raw; NBG—Nitta bean germinated; NBHTC—Nitta bean hydrothermally cooked.

**Table 4 foods-11-01822-t004:** Least gelation concentrations of Nitta bean raw, germinated, and hydrothermally cooked flours.

Sample	Concentration of Flours (g/100 mL)									
	2%	4%	6%	8%	10%	12%	14%	16%	18%	20%
NBRF	−	−	−	−	−	−	±	±	+	+
NBGF	−	−	−	−	±	+	+	+	+	+
NBHTCF	−	−	−	−	−	−	−	−	±	±

− no gelation, ± partial gelation, + complete gelation. NBRF—Nitta bean raw flour; NBGF—Nitta bean germinated flour; NBHTCF—Nitta bean hydrothermally cooked flour.

**Table 5 foods-11-01822-t005:** The functional properties of raw, germinated, and hydrothermally cooked Nitta beans flours (g 100 g^−1^).

Parameter	NBR	NBG	NBHTC
EC (%)	51.84 ± 3.74 ^b^	58.33 ± 1.67 ^b^	3.22 ± 0.06 ^a^
ES (%)	31.97 ± 7.79 ^b^	63.89 ± 2.00 ^c^	0.96 ± 0.03 ^a^
SC (mL/seed)	0.03 ±0.00 ^a^	0.12 ± 0.04 ^a^	0.11 ± 0.03 ^a^
SI	0.01±0.00 ^a^	0.02 ±0.01 ^a^	0.01± 0.00 ^a^
HC (g/seed)	0.04 ± 0.00 ^a^	0.01 ± 0.01 ^a^	0.38 ±0.00 ^b^
HI	0.53 ±0.00 ^a^	0.96 ±0.00 ^c^	0.68 ± 0.01 ^b^
WHC (mL/g)	0.81 ± 0.03 ^a^	0.82 ± 0.00 ^a^	0.83 ± 0.0 ^a^
OHC (g/g)	0.86 ± 0.03 ^a^	0.81 ± 0.03 ^a^	0.80 ± 0.05 ^a^

Results are the means of three replicates ± SE; Tukey’s HSD multiple comparison test, mean values with the same row followed by the same superscripts are not significantly different (*p* > 0.05). NBR—Nitta beans raw; NBG—Nitta beans germinated; NBHTC—Nitta beans hydrothermally cooked; EC—emulsion capacity; ES—emulsion stability; SC—swelling capacity; SS—swelling stability; HC—hydration capacity; HS—hydration stability; WHC—water-holding capacity; OHC—oil-holding capacity.

**Table 6 foods-11-01822-t006:** Functional group characteristic absorption peaks of raw, germinated, and hydrothermally cooked Nitta bean flour.

Treatments	Functional Group Characteristic Absorption Peak			
	-OH	Lipid	Protein	Carbohydrate
NBR	3404.87	3010.11, 2924.88, 2853.93, 1745.73	1643.38, 1548.1, 1398.06, 1312.64	1240.51, 1074.64
NBG	3388.28 (+)	3010.28 (+), 2925.05 (+), 2853.94 (+), 1745.77(+)	1641.27 (+), 1548.28 (+), 1455 (+), 1398.98 (+), 1309.38 (+)	1240.95 (+), 1102.61 (+), 1053.1 (−), 895.44 (−)
NBHTC	3388.28 (+)	3009.35 (+), 2924.93 (+), 2853.95 (+), 1746.29 (+)	1647.85 (+), 1542.91 (+), 1456.28 (+), 1398.09 (+), 1310.65 (+)	1239.78 (+), 1158.68, 1099.7 (+)

(+) Absorption peak was higher than that of NBR; and (−) absorption peak was lower than that of NBR. NBR—Nitta bean raw; NBG—Nitta bean germinated; NBHTC—Nitta bean hydrothermally cooked.

**Table 7 foods-11-01822-t007:** Antioxidant activity of raw, germinated and hydrothermally cooked Nitta bean flour.

Sample	ORAC (µM TE/g)	DPPH (mM TE/g)	Polyphenol (mg GE/g)
NBR	391.51 ± 4.5 ^b^	58.49 ± 3.8 ^b^	3.89 ± 0.05 ^b^
NBG	505.33 ± 4.7 ^c^	61.21 ± 3.9 ^c^	6.88 ± 0.05 ^c^
NBHTC	324.79 ± 2.9 ^a^	41.50 ± 2.1 ^a^	1.92 ± 0.03 ^a^

Results are the means of three replicates ± SE; Tukey’s HSD multiple comparison test, mean values with the same column followed by the same superscripts are not significantly different (*p* > 0.05). NBR—Nitta bean raw; NBG—Nitta bean germinated; NBHTC—Nitta bean hydrothermally cooked.

## Data Availability

The data presented in this study are available on request from the corresponding author.

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
