# Peer review of "Effect of Hydrothermal Cooking and Germination Treatment on Functional and Physicochemical Properties of Parkia timoriana Bean Flours: An Underexplored Legume Species of Parkia Genera"

_foods, 2022, doi:10.3390/foods11131822_

Round 1

Reviewer 1 Report

Please, refer to the comments given in the text of the manuscript.

Author Response

Response to Reviewer 1 Comments

We sincerely express our gratitude to the editor and the reviewers for careful and thorough reading of this manuscript and for their constructive and valuable comments which help to improve the quality of this manuscript. We have made the following revisions to justify for each comment

Point 1: Delete lines 30-33

Response: Thank you for this observation. Apology for the mistake. The correction has been made by deleting the lines 30-33.

Point 2: Inappropriate word choice, Delete supplements.

Response: The suggested correction has been made by deleting the word ‘Supplements’.

Point 3: Above it was stated that the seeds can be eaten raw or cooked with vegetables and meat. This is contradictory. Please, clarify.

Response: Thank you for the valuable comment. Nitta beans have a hard cover which makes them difficult to consume raw. Traditionally, it is eaten germinated and dehulled as well as cooked dehulled with vegetables or salad. Most of the antinutrients are present in the hard covers; therefore, soaking, germination, and cooking processes reduce the antinutrient content. The corrections have been made.

Point 4: FE-SEM or just SEM? below in R&D section FE-SEM is indicated. Please, indicate the moisture contents of the lyophilized samples prepared for SEM.

Response: Thank you for this observation. The title has been revised as FE-SEM. The moisture content of the lyophilized samples prepared for FE-SEM analysis was lower than 1% for all samples. The correction has been made.

Point 5: please, indicate the number of each method used: moisture (???), crude fat (???),etc.

Response: Thank you for this excellent observation. As per reviewer’s suggestion, details about the number of each method used to determine moisture, proteins, fats, ash, dietary fibers are mentioned. The text reads now ‘The moisture (AOAC 931.04, Oven – Memmert UN-110, Germany), crude fat (AOAC 922.06, Soxhlet – Soxtec GR-47 RA-13 006, Switzerland), ash (AOAC 930.30, Muffle Furnace – Carbolite CWF1100, UK ), crude protein (AOAC 992.23, digestion unit Buchi K435 and distillation unit Buchi B-324, Germany), total dietary fiber (AOAC 985.29, aspirator pump- Eyela A-1000S, China, oven Memmert UN-1100), and soluble (AOAC 993.19) and insoluble dietary fiber (AOAC 991.42) contents of the NBR, NBG, and NBHTC flours were estimated using the AOAC (2019) official methods [15]. The difference method was used to calculate the amount of carbohydrates in Nitta beans.

Point 6: provide the formula for calculation.

Response: Thank you for the valuable comment. The suggested correction has made by adding the formula for calculating carbohydrates as follows

Carbohydrates (%) = [100 – Moisture (%) – Proteins (%) – Fats (%) – Ash (%)]

Point 7: Please, add some details related to instrument calibration.

Response: Thank you so much for the constructive comments to improve the quality of paper. We found your comments extremely helpful and have revised accordingly. The text has been added to materials and methods of color measurement. The text now reads ‘Color analysis of the seeds and flours was conducted by color spectrometer (ColorFlex EZ, HunterLab, USA) considering different color scales (CIE L*, a*, and b*). The calorimeter was calibrated with the standard white and black plate. L* denotes lightness [from 0 (black) to 100 (white)], a* denotes reddish (+a*) and greenish (-a*) colors, and b* denotes yellowish (+b*) and bluish (-b*) colors. The measurements were performed under similar light conditions, at room temperature, replicated three times for each flour and bean sample.’

Point 8: Express in relative centrifugal force.

Response: Thank you for the valuable comment. The suggested correction has made by replacing the centrifuge speed in g force. The text reads now ‘Vortexed samples were allowed to stand for 30 min at 25 °C before being centrifuged for 25 min at 1048 g force.’ 

Point 9: It is unclear whether the results come from 3 independent biological replicates (and unknown number of repetitions) or one biological replicate in 3 repetitions. It is not the same. It would be nice to indicate total sample size

Response: Thank you for the valuable comment. We are grateful to add more details of sample size. The text has been added to materials and methods. The text now reads ‘The physicochemical parameters (n=100), proximate composition (n=3), functional properties (n=3), total polyphenol content and antioxidant activity (n=3) were performed in triplicates, and the results are presented in terms of mean ± standard error. The data were statistically analyzed using SPSS software (SPSS Inc. version 26, Chicago, IL, USA), and Tukey’s HSD multiple range test was used to determine significant differences (p < 0.05) between the mean values.’

Point 10: This is a footnote to table. Format accordingly.

Response: The suggested correction has made.

Point 11: Are data expressed in 100 g of dry matter or as is basis? It is important to indicate the basis for comparison reasons.

Response: Thank you so much for the constructive comments to improve the quality of paper. We found your comments extremely helpful and have revised accordingly. The correction has been made by expressing the data in 100 g of dry matter basis.

Point 12: proteins

Response: Thank you for this observation. Apology for the mistake. The corrections have been made by changing the word protein with ‘proteins’, fat with ‘fats’, Carbohydrate with ‘carbohydrates’, Dietary Fibers with ‘dietary fibers’.

Point 13: ???unclear. What flour? Did you mean that mechanical disintegration during milling could affect minerals or that better wash out od minerals occurred because the sample was grinded? Please, clarify.

Response: Thank you for the valuable comment. The word "flour" is not needed here. When nitta bean seeds are soaked and cooked, some water-soluble macro-nutrients and micro-nutrients that dissolve in water are washed out, which affects the mineral content. 

Point 14: p

Response: Thank you for this excellent observation. Apology for the mistake. The correction has been made by changing the letter P to p.

Point 15: Gelling ability (as stated in a prior sentence), how could it be concluded that the highest protein concentration in NBHTC is related to reduced gel-forming ability. Its contradictory. 

Response: Thank you so much for this valuable comment. Yes. I agree with the reviewer’s comment. The sentence is indeed incorrect. The statement has changed as ‘Compared with NBR, NBHTC showed the highest protein concentration, and least gel- forming ability, which is inconsistent with findings reported earlier.

Point 14: FTIR analysis not described in Materials and methods section

Response: Thank you for this excellent observation. Apology for the mistake. The correction has been made by adding FTIR analysis methodology in the methodology section. The text now reads” Fourier transform infrared (FTIR) spectra of the raw, germinated, and hydrothermally cooked Nitta bean flours were evaluated using a FTIR spectrometer (Nicolet Summit Pro, Thermo Scientific, USA) [16]. The analysis was carried out by mixing Nitta bean flour and KBr in a mass ratio of 1:100, then grinding and pressing the mixture into a pellet. Measurements were obtained in the range of 500–4000 cm−1 with a resolution of 4 cm−1, and a total of 64 scans were recorded and averaged. The experiments for FTIR were performed in triplicate”.

Point 15: Pay more attention on letter capitalization in titles. Here, antiox... 

Response: Thank you for this excellent observation. Apology for the mistake. The corrections have been made.

Reviewer 2 Report

The MS falls within scope and aims of the journal and can be on interest to wide readership of FOODS. However, I have pointed out few deficiencies for author’s consideration.

Introduction. Delete lines from 30 to 33

Common name may be added before technical name and thereafter, only common name may be used to keep uniformity.

Materials and methods

Line 144-147. To compare the concentrations of fats, ashes, proteins and fibers of raw, processed seeds and flours, I suggest standardizing the composition of each samples (raw and flours) into dry matter.

Add details about equipment/instruments like model and manufacturer etc.

Line 146: add the reference method for each determination (number of AOAC for protein ..)

Line 156: rectify the range of concentration in accord with Table 4.

In Formula (6) didn’t appear N. Please add.

Line 201: delete the word “alteration” and substitute with “ modifications”. Moreover could you explain how you obtained TPC content expressed as gallic acid equivalent.

Line 218: indicate which is the “test solution”. Line 225: rectify the statement in milligrams of Trolox equivalent per gram dry matter.

Results and “discussion” (add the word discussion)

Line 252-254: it is not necessary to repeat Table 1 in brackets.

Line 262: explain the reason for which the husks are not present in germinated and cooked samples.

Line 272-281: rectify the concentrations on dry matter (see the comment for line 144-147).

Line 489-490: TPC are expressed in mg/g in materials and methods, but in Table 7 as g/g. Please give the correct unit of measure.

Figure 3: it would be appropriate to indicate with an arrow the presence of the oval protein bodies in the cell wall on NBG flour

Author Response

Response to Reviewer 2 Comments

We sincerely express our gratitude to the editor and the reviewers for careful and thorough reading of this manuscript and for their constructive and valuable comments which help to improve the quality of this manuscript. We have made the following revisions to justify for each comment 

Point 1:  Introduction. Delete lines from 30 to 33

Response: As suggested by the reviewer, the correction has been made.

Point 2: Common name may be added before technical name and thereafter, only common name may be used to keep uniformity. 

Response: As suggested by the reviewer, the common name has been added before technical name. Further only common name is used to maintain the uniformity.

Point 3: Materials and Methods; Line 144-147. To compare the concentrations of fats, ashes, proteins and fibers of raw, processed seeds and flours, I suggest standardizing the composition of each samples (raw and flours) into dry matter. Add details about equipment/instruments like model and manufacturer etc.

Response: Thank you so much for the constructive comments to improve the quality of paper. We found your comments extremely helpful and have revised accordingly. The composition of each sample standardized into dry matter for comparing the concentrations of fats, ashes, proteins and fibers of raw and processed flour. The text has been added to materials and methods section part in proximate composition. The text now reads: “The moisture (AOAC 931.04, Oven – Memmert UN-110, Germany), crude fat (AOAC 922.06, Soxhlet – Soxtec GR-47 RA-13 006, Switzerland), ash (AOAC 930.30, Muffle Furnace – Carbolite CWF1100, UK ), crude protein (AOAC 992.23, digestion unit Buchi K435 and distillation unit Buchi B-324, Germany), total dietary fiber (AOAC 985.29, aspirator pump- Eyela A-1000S, China, oven Memmert UN-1100), and soluble (AOAC 993.19) and insoluble dietary fiber (AOAC 991.42) contents of the NBR, NBG, and NBHTC flours were estimated using the AOAC (2019) official methods [15]. The difference method was used to calculate the amount of carbohydrates in Nitta beans.”

Comment 4: Line 146: add the reference method for each determination (number of AOAC for protein ..) 

Response: As suggested by the reviewer, the reference method number for each determination has been added and the text now reads ‘The moisture (AOAC 931.04, Oven – Memmert UN-110, Germany), crude fat (AOAC 922.06, Soxhlet – Soxtec GR-47 RA-13 006, Switzerland), ash (AOAC 930.30, Muffle Furnace – Carbolite CWF1100, UK ), crude protein (AOAC 992.23, digestion unit Buchi K435 and distillation unit Buchi B-324, Germany), total dietary fiber (AOAC 985.29, aspirator pump- Eyela A-1000S, China, oven Memmert UN-1100), and soluble (AOAC 993.19) and insoluble dietary fiber (AOAC 991.42) contents of the NBR, NBG, and NBHTC flours were estimated using the AOAC (2019) official methods [15]. The difference method was used to calculate the amount of carbohydrates in Nitta beans.’

Point 5: Line 156: rectify the range of concentration in accord with Table 4. 

Response: Thank you for this comment. We agree that better use of same unit as presented in Table 4 would be accurate and have taken your advice. The correction has been made.

Point 6: In Formula (6) didn’t appear N. Please add. 

Response: Thank you for this observation. Apology for the mistake. The correction has been made in the formula. The formula now reads:

Swelling capacity (mL/seed) = (V2-V1)/N

Point 7: Line 201: delete the word “alteration” and substitute with “modifications”. Moreover, could you explain how you obtained TPC content expressed as gallic acid equivalent. 

Response: Thank you for this excellent observation. As per reviewer’s suggestion, the word “alteration” has been deleted and substituted with the word “modifications”. Gallic acid (10-80 ug/mL) was used as standard. The total phenols were expressed as mg/100 g gallic acid equivalent using the standard curve equation: y = 0.0063x + 0.0245, R2= 0.9993; Where, y is absorbance at 750 nm, and x is total phenolic content standard.

Point 8: Line 218: indicate which is the “test solution”. Line 225: rectify the statement in milligrams of Trolox equivalent per gram dry matter.

Response: Thank you for this constructive comment. In the line 218, test solution term is used for sample. The word ‘test solution’ has been replaced by the word ‘sample’. For Line 225, we do apologies for the wrongly mentioned unit of ORAC in the method, Table 7, and results and discussion section. The correct unit was micromoles of Trolox equivalent (TE) per gram sample (mM TE/g). As per reviewer’s suggestion, the statement in line 225 has been corrected and the line now reads, "The findings are presented in terms of micromoles of Trolox equivalent (TE) per gram sample (mM TE/g). Furthermore, we have made changes to the table, results and discussion. 

Point 9: Results and “discussion” (add the word discussion) 

Response: The correction has been made by adding ‘discussion’ word.

Point 10: Line 252-254: it is not necessary to repeat Table 1 in brackets 

Response: The correction has been made by deleting the word ‘Table 1’.

Point 11: Line 262: explain the reason for which the husks are not present in germinated and cooked samples.

Response: Thank you for this excellent observation. The husk content was absent in the NBG and NBHTC because it was removed during the germination process and after cooking. The corrections have been made.

Point 12: Line 272-281: rectify the concentrations on dry matter (see the comment for line 144-147). 

Response: Thank you so much for this constructive comment. The suggested correction has been made.

Point 13: Line 489-490: TPC are expressed in mg/g in materials and methods, but in Table 7 as g/g. Please give the correct unit of measure. 

Response: Thank you for this observation. Apology for the mistake. It should be mg/g instead of g/g. The correction has been made.

Point 14: Figure 3: it would be appropriate to indicate with an arrow the presence of the oval protein bodies in the cell wall on NBG flour 

Response: Thank you for the valuable comment. The correction has been made.
